# Association between Intraoperative Blood Pressure Drop and Clinically Significant Hypoperfusion in Abdominal Surgery: A Cohort Study

**DOI:** 10.3390/jcm10215010

**Published:** 2021-10-28

**Authors:** Zbigniew Putowski, Szymon Czajka, Łukasz J. Krzych

**Affiliations:** 1Students’ Scientific Society, Department of Anesthesiology and Intensive Care, School of Medicine in Katowice, Medical University of Silesia, 14 Medyków Street, 40-752 Katowice, Poland; 2Department of Anesthesiology and Intensive Care, School of Medicine in Katowice, Medical University of Silesia, 14 Medyków Street, 40-752 Katowice, Poland; szymon_czajka@wp.pl (S.C.); lkrzych@sum.edu.pl (Ł.J.K.)

**Keywords:** hypotension, intraoperative period, ischemia, postoperative complications, general surgery

## Abstract

The recent consensus by the Perioperative Quality Initiative (POQI) on intraoperative hypotension (IOH) stated that mean arterial pressure (MAP) below 60–70 mmHg is associated with myocardial infarction (MI), acute kidney injury (AKI), death and also that IOH is a function of not only severity but also of duration. However, most of the data come from large, heterogeneous cohorts of patients who underwent different surgical procedures and types of anaesthesia. We sought to assess how various definitions of IOH can predict clinically significant hypoperfusive outcomes in a homogenous cohort of generally anesthetised patients undergoing abdominal surgery, taking into account thresholds of MAP and their time durations. The data for this study come from a prospective cohort study in which patients who underwent abdominal surgery between 1 October 2018 and 15 July 2019 in the university hospital in Katowice were included in the analysis. We analysed perioperative data to assess how various IOH thresholds can predict hypoperfusive outcomes (defined as myocardial injury, acute kidney injury or stroke). 508 patients were included in the study. The total number of cases of clinically significant hypoperfusion was 38 (7.5%). We found that extending durations of low MAP, i.e., below 55 mmHg, 60 mmHg, 65 mmHg and 70 mmHg, were associated with the development of either AKI, MI or stroke. It was observed that for narrower and lower hypotension thresholds, the time required to induce complications is shorter. Patients who suffered from AKI/MI/Stroke experienced more episodes of any of the IOH definitions applied. Absolute IOH thresholds were superior to the relative definitions. For patients undergoing abdominal surgery, it is vital to prevent the extended durations of intraoperative mean arterial pressure below 70 mmHg. Finally, there appears to be no need to guide intraoperative haemodynamic therapy based on pre-induction values and, consequently, on relative drops of MAP.

## 1. Background

Over the years, there has been an increasing level of evidence linking intraoperative hypotension (IOH) to mortality, even one-year post-surgery [1]. IOH can be induced by a variety of patient- and procedure-related factors, including hypovolaemia, vasodilation and myocardial depression [2]. Additionally, ischemia-reperfusion injury due to IOH may contribute to organ damage [3]. IOH causes hypoperfusion of the vital organs such as the brain, the heart and the kidneys, leading to their dysfunction. A stroke, acute kidney injury (AKI) and myocardial infarction (MI) are the most distinct results of the clinically significant IOH. The lack of standard definitions for hypotension results in reported incidences from 5% to even 99%, depending on which definition is used and which blood pressure components are considered [4]. The recent consensus by the Perioperative Quality Initiative (POQI) on IOH stated that mean arterial pressure (MAP) below 60–70 mmHg is associated with MI, AKI, death and also that IOH is a function of not only severity but also of duration. However, these conclusions come from large cohorts of heterogenous patient populations who underwent different surgical procedures and types of anaesthesia [5,6,7,8,9,10]. Additionally, in many of these studies, IOH thresholds were pre-set by investigators. Therefore, we sought to assess how various definitions of IOH can predict clinically significant hypoperfusive outcomes in a fairly homogenous cohort of generally anesthetized patients undergoing abdominal surgery, taking into account thresholds of MAP and their time durations.

## 2. Methods

In this cohort study, we included 590 consecutive patients who underwent abdominal surgery between 1 October 2018 and 15 July 2019 in the University Clinical Center prof. K. Gibiński Silesian Medical University in Katowice. Procedures of organ procurement (*n* = 11), reoperations (*n* = 24), procedures performed under local anaesthesia or monitored anaesthetic supervision (*n* = 33), and those classified as immediate according to the NCEPOD Classification of Intervention [11] (*n* = 14) were excluded. The flow diagram for the patient selection process is presented in Figure 1. Demographic and medical data were recorded, including sex, age, weight, height, comorbidities and their pharmacological treatment, according to the ICD 10 criteria [12]. Body mass index (BMI) and Charlson Comorbidity Index (CCI) were subsequently calculated. Type and duration of anaesthesia, and type, duration and urgency of surgery were recorded [13]. The perioperative risk was assessed based on individual patients’ risk, according to the American Society of Anesthesiology (ASA) physical status (PS) classification [14], and procedural risk, according to the European Society of Cardiology and European Society of Anaesthesiology recommendations [15]. Primary arterial hypertension was diagnosed based on medical records. Currently used antihypertensive therapy was evaluated based on medical charts.

The majority of patients had their systolic (SBP) and diastolic blood pressure (DBP) measured on a non-dominant arm with an automated oscillometric non-invasive BP monitoring device (Dräger Infinity Gamma XL) with a cuff of appropriate size depending on the patient’s arm circumference and recorded in five min intervals during anaesthesia from the first pre-induction measurement until the last measurement during recovery from anaesthesia in the operating theatre. MAP values were automatically calculated. For patients with invasive blood pressure monitoring, BP was measured directly. A need for norepinephrine (NE) use, its doses and duration of infusion, together with intraoperative fluid balance, were recorded and analysed.

The following absolute MAP thresholds were distinguished: <55 mmHg, <60 mmHg, <65 mmHg, <70 mmHg, <75 mmHg. Time spent under each of described thresholds was calculated. Additionally, relative thresholds were explored, defined as a MAP drop from baseline value (pre-induction MAP), namely: >20%, >25%, >30%, >35%, >40%. Again, the duration of such IOH was also measured. The number of episodes of IOH recorded during the procedure (i.e., one 5-min interval accounted for one episode) were assessed.

The number of episodes spent under different IOH thresholds was calculated, and their association with a hypoperfusive outcome (AKI, MI or stroke) was assessed. The outcomes were defined according to their international definitions [16,17,18]. By using the ROC curve analysis of time spent under each of the IOH thresholds, we established dichotomous IOH thresholds (e.g., >5 episodes of MAP < 60 mmHg). Those definitions were then included in the multivariable logistic regression models.

STROBE (strengthening the reporting of observational studies in epidemiology) statement was applied for appropriate reporting (Appendix A) [19].

Statistical analysis was performed using MedCalc Statistical Software version 18.1 (MedCalc Software Ltd., Ostend, Belgium). Continuous variables were expressed using medians and interquartile ranges (IQR). Qualitative variables were expressed as absolute values and percentages. Between-group differences for quantitative variables were assessed using the Mann–Whitney U test or Kruskal–Wallis test. Their distribution was verified with Shapiro–Wilk test. Chi-square or Fisher’s exact test was applied for qualitative variables. Areas Under the Receiver Operating Characteristics (AUROC) curves with 95% CIs were calculated to assess the diagnostic accuracy of consecutive models. All tests were two-tailed. A “*p*” value <0.05 was considered statistically significant. All variables with significance below 0.05 in univariable testing were selected for the stepwise multivariable logistic regressions.

## 3. Results

The study group comprised 508 patients: 239 (46.6%) males and 269 (53.4%) females. The median age of patients was 62 years (IQR 46–68). Detailed study group characteristics are presented in Table 1. The median duration of anaesthesia was 230 min (IQR 130–340). There were 49 (9.6%) non-elective procedures and 245 (48.2%) oncologic surgeries. All patients received fentanyl, 94.5% of patients received propofol and 87% received sevoflurane. Anaesthesia- and surgical procedure-related variables are depicted in Table 2 and in the Appendix A.

Depending on the threshold applied, IOH could be diagnosed in 10.7–92% of all patients. The number of IOH cases increased gradually with the restriction of the IOH threshold (Table 3).

The composite primary outcome was diagnosed in 38 (7.5%) patients, including 32 cases of AKI (6.3%), 3 cases of MI (0.6%) and one event of stroke (0.2%). The association between the occurrence of a certain IOH threshold and the outcome is presented in Table 3.

In regards to the number of episodes of certain IOH thresholds, a significant difference between patients who suffered from the hypoperfusion outcomes and patients who did not experience it was observed. With each increase in the IOH threshold, the number of episodes of such hypotension increased and was greater for patients who experienced hypoperfusive outcomes (Table 4). By using the ROC curve analysis of time spent under each of the predetermined thresholds, we established dichotomous IOH thresholds (Table 4). Those definitions were then included in the multivariable logistic regression models (Table 5). Only thresholds based on absolute MAP values were statistically significant in the multivariable models.

## 4. Discussion

In this fairly homogenous abdominal surgery cohort, we found that extending durations of low MAP, i.e., below 55 mmHg, 60 mmHg, 65 mmHg and 70 mmHg, were associated with the development of either AKI, MI or stroke. There was a relationship between stricter IOH thresholds and their shorter duration required to be associated with the postoperative complications. Patients who suffered from AKI/MI/stroke experienced more episodes of any of the IOH definitions applied. Absolute IOH thresholds were superior to the relative definitions.

The results of our study are consistent with previous reports [4,10]. In a study by Walsh and Sun, any duration of MAP < 55 mmHg was associated with the development of AKI and cardiac complications [6,8]. In Walsh’s study, OR appeared much lower when assessing the risk of AKI (1.18 for 1–5 min and 1.19 for 6–10 min) and MI (1.30 for 1–5 min and 1.47 for 6–10 min). We believe such discrepancy could be a result of the difference in the study population. In our patient population, the median age was 62 years, whereas in Walsh’s cohort, it was approximately 55 years. In regards to the Charlson comorbidity index, our patients were also sicker (median 3 vs. 1 point). Considering the fraction of 3rd-grade procedural risk (25.2%) and that almost half of the procedures performed were oncological, we believe that the IOH effect on hypoperfusion in our cohort can have a substantially higher effect; therefore, higher OR occurred [15,20]. Indeed, Jang’s study, albeit performed on an orthopaedic population with a mean age of 77 years, showed that MAP < 55–60 mmHg recorded for more than 5 min (which probably corresponds to at least two episodes of MAP < 55 in our cohort) is associated with postoperative acute kidney injury with a risk of 5.14 (CI 95% 1.54–20.35, *p* = 0.012) [21].

By introducing relative IOH thresholds, we sought to explore whether relative thresholds are better suited for predicting hypoperfusive events. We found that these thresholds are not good enough to predict postoperative complications as they were not significant in multivariable models. According to the POQI statement, absolute thresholds appear to be as predictive of renal and cardiac injury as relative thresholds [4]. In our analysis, after adjustment for a number of confounding factors, only absolute thresholds were significant in relation to postoperative AKI/MI/Stroke. Based on our results, we believe that there is no need to guide intraoperative haemodynamic treatment on relative MAP drops.

As mentioned above, there was a relationship between stricter IOH thresholds and their shorter duration required to be associated with postoperative complications. Interestingly, significant IOH thresholds (Table 5) presented similar ORs and 95% CIs. We, therefore, speculate that experiencing at least one episode of MAP < 55 mmHg is somewhat equal to experiencing at least four episodes of MAP < 70 mmHg, et cetera. Those findings are in tune with several research papers and were additionally confirmed by the POQI statement that, essentially, intraoperative hypotension is a mixture of its severity and duration and is associated with AKI, MI and death [5,10]. In summary, conclusions from large, heterogeneous cohorts may be extrapolated on this homogeneous abdominal surgery population [5,6,7,8,9,10].

### Limitations

The above-mentioned findings should be interpreted with caution due to a number of possible confounding factors. Firstly, a pre-induction MAP value was defined as “baseline” MAP. It is possible that such measurement does not represent a true “baseline” as it could be influenced by stress or premedication. However, it was demonstrated that BP measurements obtained on the day of the surgery were similar to those obtained via primary care [22]. Secondly, due to the fact that BP was measured in 5-min intervals, we chose not to quantify the duration of IOH in minutes but in episodes. This, to a certain extent, limits us from assessing a relationship between IOH severity and its duration. Thirdly, the true association between IOH and hypoperfusive outcomes may be masked by the use of catecholamines. Patients who suffered from postoperative complications more often received catecholamines during the procedure. A potential generalisability of our study indeed suffers from limited study population and its homogeneity (general anaesthesia, abdominal surgery), which is, on the other hand, one of the true strengths of our research [6,19,23]. Fourthly, patients with postoperative hypoperfusion were generally sicker (had higher CCI), which is another source of bias.

The major portion of the composite outcome consisted of AKI (32 cases). This explains why chronic kidney disease was such a strong predictor of the composite outcome (Table 5 and Appendix A). Therefore, the multivariable models are greatly determined by AKI and not the other dependent variables (MI or Stroke). Therefore, we cannot imply the association between IOH and the occurrence of MI and stroke, and rather, just AKI. MI and stroke were rare, and the study was underpowered to find associations for these outcomes.

In our study, blood pressures after discharging the patient from the operating theatre were not studied. However, the role of the postoperative care period in the adverse outcomes observed cannot be underestimated. Another potential limitation is that the size of our cohort may be underpowered to detect certain associations. It is possible that, e.g., higher IOH thresholds are still associated with postoperative complications, albeit not shown directly by our study. Finally, this is an observational study, and no causal association between IOH and hypoperfusive outcomes can be demonstrated.

## 5. Conclusions

For patients undergoing abdominal surgery, it is vital to prevent the extended durations of intraoperative mean arterial pressure below 70 mmHg. There is a strong interplay between the severity and duration of intraoperative hypotension. Finally, there appears to be no need to guide intraoperative haemodynamic therapy based on pre-induction values and, consequently, on relative drops of MAP.

## Figures and Tables

**Figure 1 jcm-10-05010-f001:**
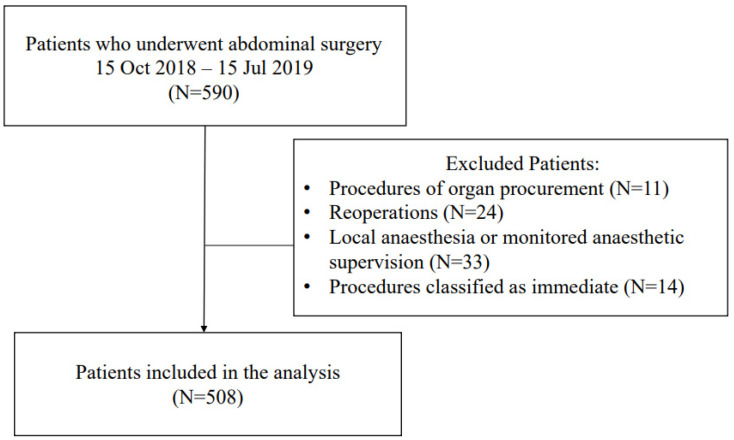
Flow diagram for the patient selection process.

**Table 1 jcm-10-05010-t001:** Preoperative population characteristics.

Variable	Outcome (−) *n* = 470	Outcome (+) *n* = 38	*p*-Value
Age (years)	61 (45–68)	67 (62–75)	0.0002
Males	219 (46.6)	20 (52.6)	0.4
BMI (kg m^−2)^	25.6 (22.5–29.0)	27.1 (21.9–29.8)	0.4
Chronic arterial hypertension	205 (43.6)	29 (76.3)	<0.0001
Chronic Kidney Disease	8 (1.7)	5 (38.5)	<0.0001
Diabetes mellitus	57 (12.1)	4 (10.5)	0.8
Pre-induction SBP (mmHg)	140 (125–153)	142.5 (130–155)	0.2
Pre-induction MAP (mmHg)	101.7 (92.0-110.0)	101.5 (95.0–110.0)	0.6
ACEI/ARB	97 (20.6)	13 (34.2)	0.05
B-blocker	118 (25.1)	15 (39.5)	0.05
Calcium antagonist	39 (8.3)	4 (10.5)	0.6
ASA PS class III/IV/V	189 (40.2)	26 (68.4)	0.0007
CCI (pts)	3 (1–5)	5 (3–7)	<0.0001
Premedication	284 (60.4)	21 (55.3)	0.5

Continuous variables are expressed as median and interquartile range (in brackets). Qualitative variables are expressed as absolute values and/or percent (in brackets).

**Table 2 jcm-10-05010-t002:** Intraoperative population characteristics.

Variable	Outcome (−) *n* = 470	Outcome (+) *n* = 38	*p*-Value
Adjunction of regional anaesthesia (number of cases)	31 (6.6)	9 (23.7)	0.0002
Invasive blood pressure monitoring (number of cases)	67 (14.7)	14 (36.8)	0.0004
Procedure Risk I *	44 (9.4)	1 (2.6)	0.1
Procedure Risk II *	314 (66.8)	21 (55.3)	0.1
Procedure Risk III *	112 (23.8)	16 (42.1)	0.01
Oncological procedure (number of cases)	219 (46.6)	26 (68.4)	0.009
Catecholamine use (number of cases)	197 (41.9)	30 (78.9)	<0.0001
Time of catecholamine administration from the induction of anaesthesia (min)	40.0 (20.0–80.0)	37.5 (15.0–60.0)	0.4
Catecholamine dose (µg kg^−1^ min^−1^)	0.054 (0.042–0.090)	0.070 (0.048–0.091)	0.3
Procedure duration (min)	215.0 (120.0–330.0)	372.5 (235.0–492.0)	<0.0001
Fluid dose (mL kg^−1^ h^−1^)	6.78 (5.16–8.76)	6.67 4.74–8.58)	0.5

Continuous variables are expressed as median and interquartile range (in brackets). Qualitative variables are expressed as absolute values and/or percent (in brackets). * according to European Society of Cardiology and European Society of Anaesthesiology recommendations [15].

**Table 3 jcm-10-05010-t003:** Absolute and relative MAP thresholds and their association with the negative outcome.

Threshold	Outcome (−) *n* = 470	Outcome (+) *n* = 38	*p*-Value
<55 mmHg	44 (9.4)	10 (26.3)	0.001
<60 mmHg	107 (22.9)	17 (44.7)	0.002
<65 mmHg	218 (46.4)	26 (68.4)	0.009
<70 mmHg	345 (73.4)	29 (76.3)	0.6
<75 mmHg	412 (88.0)	35 (92.1)	0.4
Drop > 20% from baseline	432 (91.9)	35 (92.1)	0.9
Drop > 25% from baseline	391 (83.2)	33 (86.8)	0.5
Drop > 30% from baseline	319 (67.9)	31 (81.6)	0.07
Drop > 35% from baseline	243 (51.7)	25 (65.8)	0.09
Drop > 40% from baseline	162 (34.5)	23 (60.5)	0.001

Qualitative variables are expressed as absolute values and/or percent (in brackets).

**Table 4 jcm-10-05010-t004:** Number of episodes spent under each of absolute and relative IOH thresholds.

Threshold	Outcome (−) *n* = 470	Outcome (+) *n* = 38	Best Cut-Off Threshold in Predicting Outcome (+)	*p*-Value
<55 mmHg	0 (0–0)	0 (0–1)	>0	0.001
<60 mmHg	0 (0–0)	0 (0–2)	>0	0.0009
<65 mmHg	0 (0–2)	2 (0–3)	>1	0.003
<70 mmHg	2 (0–6)	6 (1–16)	>4	0.01
<75 mmHg	7 (2–13)	13 (3–29)	>17	0.005
Drop > 20% from baseline	15 (5–32)	29 (9–55)	>27	0.006
Drop > 25% from baseline	9 (1–19)	16 (3–36)	>20	0.04
Drop > 30% from baseline	3 (0–10)	5 (1–19)	>2	0.03
Drop > 35% from baseline	1 (0–3)	2 (0–6)	>4	0.04
Drop > 40% from baseline	0 (0–1)	1 (0–3)	>0	0.001

Continuous variables are expressed as median and interquartile range (in brackets). Cut-off points for number of IOH episodes were calculated by using ROC-curve analysis.

**Table 5 jcm-10-05010-t005:** Multivariable logistic regression models for prediction of the hypoperfusion-related events. Adjusted for chronic kidney disease, procedure duration and arterial hypertension (Appendix A).

Model	OR (95% CI)	*p*-Value
>0 episodes of MAP < 55 mmHg	2.56 (1.05–6.26)	0.039
>0 episodes of MAP < 60 mmHg	2.61 (1.22–5.59)	0.01
>1 episodes of MAP < 65 mmHg	2.50 (1.17–5.30)	0.017
>4 episodes of MAP < 70 mmHg	2.67 (1.26–5.67)	0.01
>17 episodes of MAP < 75 mmHg	*	*
>27 episodes of MAP drop > 20% from baseline	*	*
>20 episodes of MAP drop > 25% from baseline	*	*
>2 episodes of MAP drop > 30% from baseline	*	*
>4 episodes of MAP drop > 35% from baseline	*	*
>0 episodes of MAP drop > 40% from baseline	*	*

Other variables included in the models were as follows: Chronic Kindey Disease (1/0), duration of the procedure (per 1 min), arterial hypertension (1/0). “*” IOH thresholds that failed to be included in the multivariable models.

## Data Availability

The datasets used and/or analysed during the current study are available from the corresponding author on reasonable request.

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
