# Peer review of "Association between Intraoperative Blood Pressure Drop and Clinically Significant Hypoperfusion in Abdominal Surgery: A Cohort Study"

_jcm, 2021, doi:10.3390/jcm10215010_

Round 1

Reviewer 1 Report

Thank you for your contribution.
A limitation is that the number of enrolled patients is small, and it is worth mentioning the process of calculating the number.
In the reference format, except for number 18, numbering is omitted, so please correct it according to the submission format. The topic and results are interesting. Thank you.

Author Response

Dear reviewer,

Thank you for your comments. We have addressed them fully. We have added new limitations regarding the power of the study. Additionally, we have fixed the reference format.

Reviewer 2 Report

  1. Please avoid abbreviations without their explanation, when used first (e.g. line 14). Use only abbreviations after that.
  2. Line 15: (IOH is a function of not only severity, but also of duration). I would recommend the extent of adverse outcome depends on severity of hypotension and its duration.  
  3. Line 16: add cohorts of patients…
  4. Line 18: use patients undergoing abdominal surgery (here and within the text).
  5. Lines 24-25: (Total number of cases of clinically significant hypoperfusion was 38). Arterial hypotension (or hypotensive episodes) is the correct term to use. If you use hypoperfusion, you should define the hypoperfused organs. If it is systemic hypoperfusion of tissues, then it is called shock.
  6. Lines 25-26: (We found that extending durations of low MAP, i.e. be- 25 low 55 mmHg, 60 mmHg, 65 mmHg and 70 mmHg were associated with the development of either 26 AKI, MI or stroke). MAP 65-70 is not hypotension, and it is not associated with adverse effects. Please clarify. Also, consider the potential hazards associated with arterial hypertension and the methods to maintain high blood pressure.
  7. Lines 27-28: Make the sentence grammatically correct.
  8. Lines 30-32: Modify “abdominal surgery patients”, use extended instead of extending.
  9. Line 32: Sentence needs modification or clarification.
  10. Line 40 Replace “affected” with “induced” or precipitated”.
  11. Line 59, abstract, and figure 1. Check and correct the numbers of patients.
  12. Line 60, name the hospital.
  13. Line 61: use under local anesthesia.
  14. Line 66: use their instead of its.
  15. Line 85: use mean BP thresholds.
  16. Power of the study is not assessed or presented. If data are underpowered (which may be the case with relatively small number of patients), that should be stated and the results presented as trends suggesting that the hypothesis may be right.
  17. The word outcome is not explanatory enough in the tables. Define what it means. The tables must be self-explanatory.
  18. Even though the authors describe the cohort as homogenous, there is insufficient information to support that statement. There is little information about the types of surgery and coexisting conditions (diabetes, etc.).
  19. The study seems underpowered, and the results should be interpreted with caution. Such factors as hydration, intraoperative blood loss and requirement for transfusion, and many others could have influenced the results. The authors do not present enough information about all these additional factors which could contribute to adverse outcome in some patients.

Author Response

Dear Reviewer,
Thank you for reading our manuscript and for your valuable comments. We are pleased to answer
them as follows:
1) All abbreviations were explained when used for the first time. Thank you.
2) Line 15 – corrected as proposed.
3) Line 16 – corrected as proposed.
4) Line 18 – corrected as proposed.
5) Line 24-25 - we defined our outcome as adverse events following surgical procedures which
may result from hypotensive episodes caused by hypotension episodes occurring during
anesthesia. It is, therefore, correct in our opinion to use the term “hypoperfusive events”
(MI, AKI, stroke) in the abovementioned context.
6) Lines 27-28: Our correction: “It was observed that for narrower and lower hypotension
thresholds, the time required to induce complications is shorter.”
7) Lines 30-32: corrected, thank you.
8) Line 32: we deleted that sentence
9) Line 40 Replaced as proposed, thank you

10) Line 59, abstract: corrected. 590 is correct.
11) Line 60– corrected as proposed.
12) Line 61– corrected as proposed.
13) Line 66– corrected as proposed.
14) Line 85: – corrected as proposed.
15) We did not perform power calculation as we were only investigating the association between
IOH and postoperative outcomes in an observational manner. We are aware that our study
might be underpowered to detect certain associations and we stated this in the “limitation”
section.
16) Tables were corrected as requested
17) In order to address this comment we decided to add another “Types of surgery” table to the Supplementary material. Additionally, all of our patients underwent general anaesthesia, underwent abdominal surgery, received propofol, sevoflurane and fentanyl.
to our manuscript.
18) Information concerning fluid therapy was included in the text.
19) Fluid therapy was only used to correct intraoperative fluid loss. We have added information
about possible potential confounders to our observations to the limitations section.

Round 2

Reviewer 2 Report

  1. Again, once you have defined the abbreviation in the text, use only the abbreviations thereafter (e.g., line 58). There is no need to define the abbreviations every time.
  2. According to tables 1 and 2, the demographic and preexisting parameters, as well as the duration of surgery are different in patient groups, and this is an important contributor to differences in outcomes. The authors have mentioned this in their limitations. I would recommend adding a sentence in the discussion as well to highlight this differences and their potential association with the outcomes.
  3. Table 2  indicated that invasive blood pressure was monitored in many patients. Please add a clarification in the methods explaining which blood pressure was used for calculations for those patients (i.e. the described oscillotonometric method or the direct pressure).

Author Response

Dear reviewer,

Thank you for your input,

  1. We have corrected the abbreviations.
  2. We are aware of the effect that various demographic and perioperative variables can have on the outcomes. This is exactly why we have performed multivariable logistic regressions. In our analyses, chronic kidney disease, duration of the procedure, IOH and arterial hypertension contributed to the postoperative outcomes in an independent fashion. This is why we believe it is not necessary to mention the possible confounding effect of those variables in the limitations section.
  3. We added an explanation regarding an invasive blood pressure monitoring in the methods section.

Thank you and best regards